# XES3G5M: A Knowledge Tracing Benchmark Dataset with Auxiliary Information

**Zitao Liu**
Guangdong Institute of Smart
Education, Jinan University
Guangzhou, China
liuzitao@jnu.edu.cn

**Qiongqiong Liu**
TAL Education Group
Beijing, China
liuqiongqiong1@tal.com

**Teng Guo**[*]
Guangdong Institute of Smart
Education, Jinan University
Guangzhou, China
teng.guo@outlook.com

**Jiahao Chen**
TAL Education Group
Beijing, China
chenjiahao@tal.com

**Shuyan Huang**
TAL Education Group
Beijing, China
huangshuyan@tal.com

**Xiangyu Zhao**
School of Data Science
City University of Hong Kong
Hong Kong, China
xianzhao@cityu.edu.hk

**Jiliang Tang**
Department of Computer Science
and Engineering, Michigan
State University MI, USA
tangjili@msu.edu

**Weiqi Luo**
Guangdong Institute of Smart
Education, Jinan University
Guangzhou, China
lwq@jnu.edu.cn

**Jian Weng**
College of Cyber Security
Jinan University
Guangzhou, China
cryptjweng@gmail.com

## Abstract

Knowledge tracing (KT) is a task that predicts students' future performance based on their historical learning interactions. With the rapid development of deep learning techniques, existing KT approaches follow a data-driven paradigm that uses massive problem-solving records to model students' learning processes. However, although the educational contexts contain various factors that may have an influence on student learning outcomes, existing public KT datasets mainly consist of anonymized ID-like features, which may hinder the research advances towards this field. Therefore, in this work, we present, *XES3G5M*, a large-scale dataset with rich auxiliary information about questions and their associated knowledge components (KCs)[2]. The XES3G5M dataset is collected from a real-world online math learning platform, which contains 7,652 questions, and 865 KCs with 5,549,635 interactions from 18,066 students. To the best of our knowledge, the XES3G5M dataset not only has the largest number of KCs in math domain but contains the richest contextual information including tree structured KC relations, question types, textual contents and analysis and student response timestamps. Furthermore, we build a comprehensive benchmark on 19 state-of-the-art deep learning based knowledge tracing (DLKT) models. Extensive experiments demonstrate the effectiveness of leveraging the auxiliary information in our XES3G5M with DLKT models. We hope the proposed dataset can effectively facilitate the KT research work.

---

[*]The corresponding authors: Teng Guo, Qiongqiong Liu.

[2]A KC is a generalization of everyday terms like concept, principle, fact, or skill.

# 1 Introduction

Knowledge tracing (KT) is the task of *using students' historical learning interaction data to model their knowledge mastery over time so as to make predictions on their future interaction performance* (shown in Figure 1). Solving the KT problems may help teachers better detect students that need further attention, or recommend personalized learning materials to students which is essential for building next-generation smart and personalized education. Recently, many KT studies propose deep learning based KT (DLKT) models following the data-driven paradigm that uses lots of students' historical interaction sequences to estimate their knowledge states which have achieved promising results [25, 36, 22, 7, 18].

However, the success of existing DLKT research is heavily relied on how much information is revealed from the third-party education platforms due to the special characteristics of educational data such as data privacy. The majority of the widely used KT datasets such as ASSISTments [25], Peking Online Judge [23] and KDD datasets [7] only contain ID features of questions and knowledge components (KCs), which lack neces-

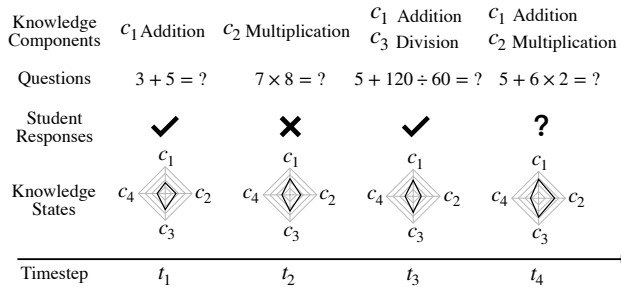

Figure 1: The graphical illustration of the KT problem.

sary contextual information of questions and KCs. A KC is a description of a mental structure or process that a learner uses, alone or in combination with other KCs, to accomplish steps in a task or a problem. Furthermore, the existing KT datasets in math domain only cover a small number of KCs, which may not be able to represent the entire knowledge states even for students in a particular grade.

Therefore, in this work, we propose a large-scale dataset named XES3G5M which consists of student interaction logs collected from a K-12 online learning platform in China. Specifically, the XES3G5M dataset is made up of 7,652 questions, 865 KCs and 5,549,635 interactions from 18,066 students. The XES3G5M provides extremely rich auxiliary information from diverse educational scenes that can be summarized as follows:

- **Question side information**: the dataset provides plenty of question-side information including question textual content, question types, and answer analysis which provides problem-solving ideas. The information on the question side is provided in detail and is helpful to discriminate the questions with the same KC sets.
- **KC side information**: the KCs in XES3G5M are in a multi-level structure and each KC has a corresponding route to indicate its relation with other KCs, which can be used to capture the latent dependency of KCs.

The KC route is presented as a structural tree that explicitly introduces the prerequisite relations among all the 865 KCs involved in the proposed dataset. To the best of our knowledge, our XES3G5M dataset not only has the largest number of KCs in math domain but contains the richest contextual information including tree structured KC relations, question types, textual contents and analysis and student response timestamps. Meanwhile, we present a systematic KT benchmark on XES3G5M with 19 state-of-the-art DLKT models to evaluate the applicability and effectiveness of our dataset. We further conduct comprehensive experimental analysis to demonstrate the improvements of leveraging the auxiliary information in DLKT models. Moreover, we also conduct multi-step ahead predictions that predict students' future responses given the limited historical interactions to meet the real-world educational scenario.

# 2 Related Work

## 2.1 Knowledge Tracing Datasets

Recently, there are lots of educational datasets that involve students' question-solving logs that have been widely used in KT task. For example, ASSISTments datasets like ASSISTments2009, ASSISTments2015 are collected from ASSISTments online learning platform, which supplies instructional

counseling by measuring the students' knowledge states [6, 24]. Junyi Academy is an online learning activity dataset that mainly solves 722 mathematics questions from an online learning platform in Taiwan[3]. Peking Online Judge (POJ) is derived from the Peking online platform, which offers students coding practices to help them improve their programming skills, and is made available by [23]. EdNet contains 784,309 students' learning activities for preparing the TOEIC (Test of English for International Communication®) test from an AI tutoring service platform named Santa in South Korea [5]. The Statics2011 dataset contains 333 student interactions from an online engineering statics educational course at Carnegie Mellon University during Fall 2011[4].

Meanwhile, there are some high-quality students' learning sequences data released by various top data mining competitions. KDD datasets stem from the KDD Cup 2010 EDM Challenge which includes the Algebra question-solving logs of 13-14 years old students with detailed step-level student responses[5]. The NeurIPS2020 dataset is provided by the NeurIPS2020 Education Challenge. The recordings for Tasks 3 and 4 in NeurIPS2020 consist of student responses to mathematical questions from Eedi, which engages millions of students in daily interactions around the world[6].

However, as the comparison results are shown in Table 1, all aforementioned datasets mainly contain the ID features of questions and KCs and timestamps but lack information related to educational contexts that may potentially be useful in assessing students' knowledge states [23, 31, 32, 7]. Compared to these datasets, our XES3G5M has extra question textual contents, KC relationships, question types, and answer analysis that may be utilized to enhance the modeling process of the student's learning outcomes.

Table 1: Comparisons between existing benchmark KT datasets and XES3G5M.

| | ASSISTments | | | | Statics | Junyi | KDD | | NeurIPS | POJ | EdNet | XES3G5M |
|---|---|---|---|---|---|---|---|---|---|---|---|---|
| | 2009 | 2012 | 2015 | 2017 | 2011 | 2015 | 2005 | 2006 | 2020 | | | |
| # of Students | 4,217 | 46,674 | 19,917 | 1,709 | 333 | 247,606 | 574 | 1,146 | 4,918 | 22,916 | 1,677,583 | 18,066 |
| # of Ques. | 26,688 | 179,999 | 100 | 3,162 | 1,224 | 722 | 210,710 | 207,856 | 948 | 2,750 | 52,676 | 7,652 |
| # of KCs | 123 | 265 | - | 102 | - | 41 | 112 | 493 | 57 | - | 962 | 865 |
| # of Interactions | 346,860 | 6,123,270 | 708,631 | 942,816 | 194,947 | 25,925,922 | 809,694 | 3,679,199 | 1,382,727 | 996,240 | 372,366,720 | 5,549,635 |
| Subject | Math | Math | Math | Math | Math | Math | Math | Math | Math | PL | Linguistics | Math |
| Language | English | English | English | English | English | Chinese | English | English | English | English | English | Chinese |
| Timestamp avail. | No | Yes | No | Yes | Yes | Yes | Yes | Yes | Yes | Yes | Yes | Yes |
| Contents avail. | No | No | No | No | No | Yes | No | No | No | No | No | Yes |
| KC relation avail. | No | No | No | No | No | No | No | No | No | No | No | Yes |
| Ques. type avail. | No | No | No | No | No | No | No | No | No | No | No | Yes |
| Ques. analysis avail. | No | No | No | No | No | No | No | No | No | No | No | Yes |

## 2.2 Deep Learning Based KT Models

Many KT literature apply deep learning techniques to solve the KT problem. These DLKT models can be categorized into 5 categories as follows:

**C1: Deep sequential models**. The models capture the students' knowledge states by using auto-regressive architectures to model the historical learning interactions [3, 8, 11, 12, 19, 20, 25, 31, 35, 30, 28, 18]. For example, Piech et al. exploited an LSTM layer to estimate the student knowledge states to predict their response performances [25].

**C2: Memory augmented models**. The models use external memory networks to capture the latent relations between KCs to predict students' knowledge states [1, 28, 36, 34]. For example, Zhang et al. used a static key memory matrix to model the KC latent relationships and predicted the students' knowledge mastery levels via a dynamic value memory matrix [36].

**C3: Adversarial based models**. The models use adversarial training techniques to improve their generalization capability by performing adversarial perturbations to the original student interaction sequences [8]. For example, Guo et al. generated adversarial perturbations into interaction sequences to reduce the risk of the overfitting and limited generalization problem of DLKT models [8].

**C4: Graph based models**. The models use graph neural networks to model intrinsic relations among questions, KCs and interactions [21, 32, 33]. For example, Liu et al. captured the question-level and KC-level inner relations via a question-KC bipartite graph to augment the question and KC representations of DLKT model [14].

---

[3]https://www.junyiacademy.org/
[4]https://pslcdatashop.web.cmu.edu/DatasetInfo?datasetId=507
[5]https://kdd.org/kdd-cup/view/kdd-cup-2010-student-performance-evaluation/Data
[6]https://eedi.com/projects/neurips-education-challenge

**C5: Attention based models**. The models use attention mechanisms to capture the relevance between historical interactions and the future questions [7, 23, 26, 37, 22, 4]. For example, Pandey & Karypis leveraged a self-attention network to estimate the relation between exercises and responses [22].

Although the aforementioned DLKT approaches have achieved remarkable results, [7, 21, 25, 29, 36], recent studies [11, 12, 25, 31] seem to resemble each other with very limited nuances from the methodological perspective. Most of the existing work only provides coarse evaluation and both the contributing factors leading to the success of DLKT and how the DLKT models perform in real-world educational contexts still remain somewhat unknown. Therefore, in this work, we conduct standardized evaluations with 19 classical DLKT models on the proposed XES3G5M dataset, which ensures that all the models can be compared in a fair evaluation protocol. With the proposed benchmark, the KT researchers are able to evaluate their proposed approaches against a wide range of advanced methods on the XES3G5M dataset and the practitioners are capable of discriminating against the opportunities and challenges of the DLKT algorithms in real-world educational contexts.

## 3 Data Description

### 3.1 License

The dataset can be freely downloaded from `https://github.com/ai4ed/XES3G5M` and used with the MIT license[7]. Besides the KT task, the users can use the dataset for other custom tasks such as exercise recommendation under the license.

### 3.2 Data Collection and Preprocessing

To construct a high-quality educational dataset containing rich auxiliary information, we mainly collect data from two perspectives, including questions designed by teachers during class and homework assignments. First, for the questions in class, we collect educational data from an online learning system that is developed by TAL Education Group (TAL), one of the largest educational technology companies in China. For each sample, we collect question textual information and the corresponding student's answer. Second, for the questions of homework assignments, we utilize a series of auxiliary educational tools of TAL such as personalized learning applications to collect students' question-solving logs in their homework.

To this end, the dataset includes more than 5 million interactions collected from more than 18,000 third-grade students that respond to about 8,000 math questions. In XES3G5M, we only select the students' sequences with more than 200 interactions to better model students' learning processes with enough historical interaction information. Furthermore, we perform data preprocessing with the following steps to obtain the XES3G5M dataset: (1) we remove the questions with missing and incomplete content; (2) we remove the questions that miss the associated KCs; (3) the questions in the dataset are only in single choice and fill-in-the-blank types; (4) to evaluate the 19 DLKT models under a rigorous evaluation protocol, we further conduct data preprocessing following [17]. The data statistics after data preprocessing is described in Section 3.5.

### 3.3 Privacy

In this dataset, we have collected students' interaction sequences, including student IDs, question IDs, answers (correct or incorrect), timestamps, question contents, KCs associated with the questions, and question types. Among the selected information, the student IDs and question IDs have the potential to reveal the identity of students and questions. Therefore, to better protect students' privacy, we have employed a lightweight data encryption method based on data mapping. For each student ID and question ID, we map it into a non-reversible digital identifier. The mapping identifier generates unique IDs in the dataset to guarantee each student and question has a unique identifier to avoid any duplication or confusion. The encryption method enables data security meanwhile improving data processing efficiency.

---

[7]https://github.com/ai4ed/XES3G5M/blob/main/LICENSE

### 3.4 Interactions with Auxiliary Information

As shown in Table 1, XES3G5M has plentiful auxiliary information compared to existing educational question-solving logs datasets. In the following subsections, we will introduce the interaction sequences information and the corresponding auxiliary information in detail.

#### 3.4.1 Interactions

Students' learning interactions are crucial to model their learning processes in the KT problem. In XES3G5M, each interaction has the following information: (1) **question ID**: the unique question ID that the student answered; (2) **KC ID**: the corresponding leaf KC IDs associated with the answered question; (3) **response**: the correctness of the response of a student, 1 denotes correct and 0 denotes incorrect; (4) **timestamp**: the start timestamp of the student answering the question.

#### 3.4.2 Auxiliary Information

Besides the interaction information mentioned above, we collect additional auxiliary information to provide more fine-grained information to enhance a student's knowledge mastery estimation. Specifically, the auxiliary information is included in five types:

- **Question Content**: question textual content and the associated images. The examples of multi-choice questions and fill-in-the-blank questions are illustrated in Figure 2. More examples are shown in Appendix A.4. It can be seen that the image provides indispensable information for understanding multi-choice questions shown on the left of Figure 2 that are neglected in existing datasets, which is helpful to enhance the question representation during the interaction encoding. The original question textual content is stored in Chinese and we open sourced both the original Chinese version and the numerical embeddings we extract from an in-house pre-trained large-scale language model built upon math related corpus.

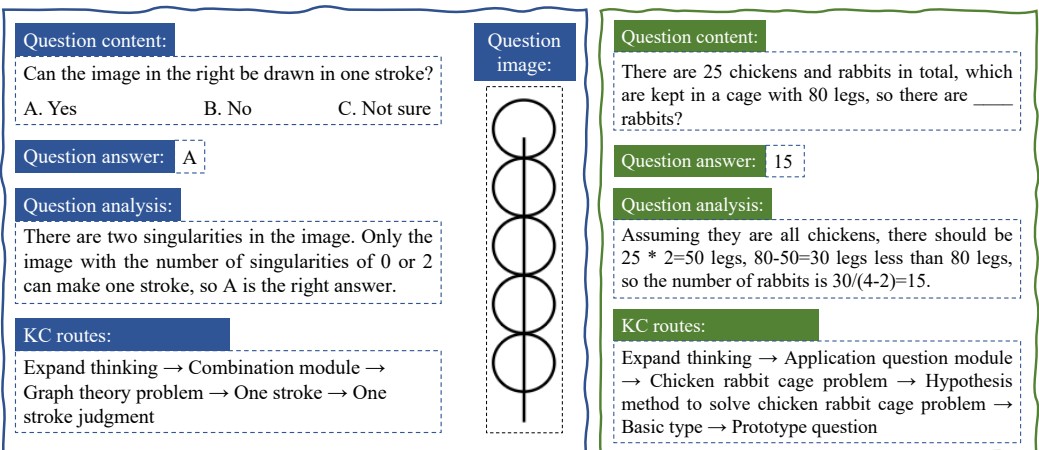

Figure 2: The examples of a multi-choice question and a fill-in-the-blank question in XES3G5M (questions have been translated).

- **Question Answer**: the standardized correct answer to the question is included in our dataset. We provide the correct options for multi-choice questions and the correct answer contents for fill-in-the-blank questions.

- **Question Analysis**: the detailed process of solving the question, including the principles and the particular steps required to solve the questions as shown in Figure 2.

- **KC Routes**: the relation among KCs. Practically, the KCs are subordinate to each other rather than independent. All the KCs in our dataset are formed hierarchically in a tree structure, and a KC route is a path from the root node of a KC to a leaf node of a KC in the tree. Since we construct KC routes from different perspectives, different KC routes may have the same leaf node. Due to the space limit, the example of KT routes is shown in Appendix A.5. With the KC routes in the dataset, we can obtain more informational representations of questions and KCs, yielding measure the relevance between questions and responses.

- **Question Type**: the type of question. The question type of questions in our dataset contains multi-choice and fill-in-the-blank questions.

Due to the space limit, the dataset structure and storage format are described in Appendix A.6.

### 3.5 Statistics

Table 1 shows the statistics of the basic information for XES3G5M. There are 5,549,635 interactions and about 18,066 students answered 7,652 questions from 865 leaf-node KCs. The average number of KCs for each question is 1.1640. There are 6,142 and 1,510 questions for fill-in-the-blank and multi-choice questions respectively. The 82.98% questions and 92.37% KCs in the dataset have more than 20 interactions to be answered by students.

Figure 3 shows the distributions of students' interaction sequence information in XES3G5M. Figure 3(a) shows the distribution of students' interaction sequence length. From the Figure 3(a), we can see that there are 89.07% student sequences containing 200 to 500 interactions. From figure 3(b), we can see that more than half of the students in the dataset (54.42%) can answer up to 80% questions correctly among all the questions they answered. Figure 3(c) and (d) illustrate the temporal information analysis for the students' sequences. We calculate the average variance of the time span between two consecutive interactions in each student's interaction sequence. From Figure 3(c), we observe the average variance of the time span between two consecutive interactions is always 80 hours. This indicates there is a long interval time between students' consecutive interactions. We also compute the whole duration of each student responding to all the questions of a learning sequence, as shown in Figure 3(d). It can be seen that almost all students in the dataset interact with questions for more than 1 year. Due to the space limit, the distributions of questions' information are shown in Appendix A.7.

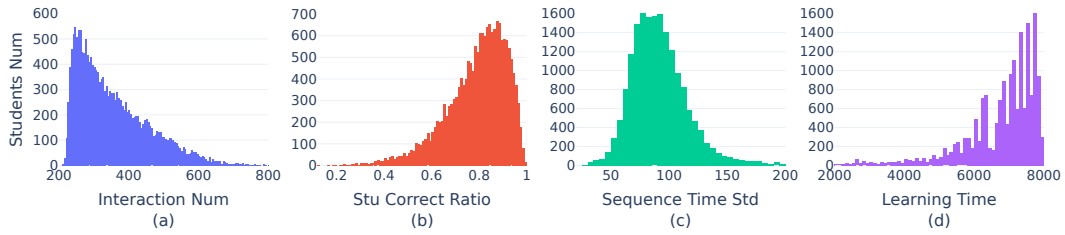

Figure 3: The distributions about interaction length, correct answer ratio, the average variance of the time span between two consecutive interactions and the whole duration of each student's sequence.

## 4 The XES3G5M Benchmark

### 4.1 Experimental Setup

In this section, we conduct the classical DLKT models on the proposed XES3G5M. We perform standard 5-fold cross validation for all the combinations of pairs of method and dataset. We tune the parameters on the validation set and report the performance on test set. The embedding size of all the models is set to [64, 128]. We set the learning rate as [1e-3, 1e-4, 1e-5] and the dropout rate is set to [0.05, 0.1, 0.3, 0.5]. We adopt the Adam [10] optimizer to train all models for 200 epochs. We stop the training process if the AUC scores do not improve in 10 epochs. We run all the models on NVIDIA RTX A5000 GPU devices with Pytorch. For the evaluation metric, we choose the Area Under the Curve (AUC) and accuracy to evaluate the prediction performance similar to previous DLKT studies [25, 35, 20, 30, 36, 1, 34, 8, 21, 22, 4, 7, 28, 18].

### 4.2 Baselines

In our benchmark, we mainly select 19 state-of-the-art DLKT models in 5 categories discussed in Section 2.2 as follows:

- **C1: DKT** [25]: it uses an LSTM layer to model students' learning processes.

- **C1: DKT+** [35]: it is a variant of DKT to address the reconstruction and inconsistent issues in the original DKT model.
- **C1: DKT-F** [20]: it is also an extension of DKT that considers students' forgetting behaviors to predict student performance.
- **C1: KQN** [11]: it is an RNN-based model that extracts the relation representation between students' learning abilities and KCs to predict students' performance.
- **C1: qDKT** [30]: it is a variant of DKT that models each learner's response performance on individual questions over time.
- **C1: IEKT** [18]: it estimates students' knowledge states via the student cognition and knowledge acquisition estimation modules.
- **C1: DIMKT** [27]: it considers personalized knowledge acquisition of a student and dynamically estimates the student's knowledge states by exploring the question difficulty effect.
- **C1: AT-DKT** [15]: it performs two auxiliary learning tasks including question tagging prediction and individualized prior knowledge prediction task to enhance the predictive capability of DKT.
- **C1: QIKT** [2]: it is a question-centric interpretable KT model jointly learned from a question-centric knowledge acquisition module and a question-centric problem-solving module.
- **C2: DKVMN** [36]: it exploits two memory networks to calculate the relationships between potential concepts and directly predicts a student's knowledge state of each concept respectively.
- **C2: SKVMN** [1]: it is a combination of DKVMN and LSTM that uses a hop-LSTM layer to capture sequential dependencies of exercises.
- **C2: DeepIRT** [34]: it incorporates DKVMN and item response theory to enhance the interpretability of the prediction output of DKVMN.
- **C3: ATKT** [8]: it improves the generalization of the attention-LSTM based KT model via adversarial training.
- **C4: GKT** [21]: it casts the knowledge structure as a graph and reformulates the KT task as a time series node-level classification problem via a graph neural network.
- **C5: SAKT** [22]: it uses self-attention mechanism to measure the relevance between the current KCs and the students' historical interactions.
- **C5: SAINT** [4]: it is a Transformer-based model that encodes exercise information and historical responses in the encoder and decoder respectively to estimate students' current responses.
- **C5: AKT** [7]: it uses a monotonic attention mechanism to identify the relevance of questions and responses and uses a Rasch model to capture the relation representations of KCs and questions.
- **C5: simpleKT** [16]: this is a strong but simple baseline method that models question-specific variations to capture the individual differences among questions covering the same set of KCs.
- **C5: sparseKT** [9]: it incorporates a k-selection module to select relevant historical interactions with the highest attention scores to improve the robustness and generalization of the attention based DLKT models.

### 4.3 Performance Analysis only with ID Features

Table 2 shows the performance of all DLKT models on XES3G5M. We report the average scores of AUC and accuracy and the standard deviations (SD) across 5 folds. As we can see that: (1) the DLKT models which integrate the KC representations and question representations together such as IEKT and QIKT mostly outperform other models which only extract KCs information like DKT, DKT+, KQN, DKMVN, ATKT, GKT, SAKT, SKVMN in terms of AUC scores and accuracy. This indicates that the questions and KCs contained in our proposed dataset are both beneficial to predict the students' future performance; (2) in particular, IEKT achieves the best AUC and accuracy compared to other KT models. We believe that it is due to IEKT extra uses historical prediction results when predicting the current response of students; (3) compared to DKT, DKT-F extracts temporal information to model the students' forgetting behavior when estimating student's knowledge states which indicate the effectiveness of temporal features like timestamps in the KT datasets; (4) among attention based DLKT models including SAKT, SAINT, AKT, simpleKT and sparseKT, AKT performs achieves the best AUC and accuracy. This may be due to AKT considering the student's

learning behavior via a monotonic attention module. In general, our XES3G5M contains diverse features of students' interaction logs which can be widely used in various KT models.

## 4.4 Performance Analysis with Auxiliary Information

We conduct additional experiments of leveraging the auxiliary information on KT models to examine the effectiveness of augmenting KT baselines with the auxiliary information in our XES3G5M dataset. Specifically, there are three different methods for the fusion of question and KC information: (1) **Add**: the embeddings of question information and KC information in XES3G5M are combined together and added to the original embedding of question ID and KC ID respectively. (2) **Early Fusion**: keep the original embedding of question ID and KC ID unchanged. The other information of the question and KCs in XES3G5M are merged by a gate unit before as the input embedding to DLKT models. (3) **Late Fusion**: the additional information of the question and KCs is combined via a gate unit after the output of DLKT models.

Table 2: The AUC and accuracy performance of KT baselines on XES3G5M dataset.

| Model | Ques. ID | KC ID | Timestamp | AUC | Accuracy |
|---|---|---|---|---|---|
| DKT | No | Yes | No | 0.7852±0.0006 | 0.8173±0.0002 |
| DKT+ | No | Yes | No | 0.7861±0.0002 | 0.8178±0.0001 |
| DKT-F | No | Yes | Yes | 0.7940±0.0006 | 0.8209±0.0003 |
| KQN | No | Yes | No | 0.7793±0.0006 | 0.8152±0.0002 |
| qDKT | Yes | No | No | 0.8225±0.0002 | 0.8301±0.0000 |
| IEKT | Yes | Yes | No | 0.8280±0.0002 | 0.8316±0.0002 |
| DIMKT | Yes | Yes | No | 0.8220±0.0002 | 0.8291±0.0006 |
| AT-DKT | Yes | Yes | No | 0.7932±0.0004 | 0.8198±0.0004 |
| QIKT | Yes | Yes | No | 0.8222±0.0006 | 0.8300±0.0005 |
| DKVMN | No | Yes | No | 0.7792±0.0004 | 0.8155±0.0001 |
| SKVMN | No | Yes | No | 0.7514±0.0005 | 0.8075±0.0003 |
| DeepIRT | No | Yes | No | 0.7785±0.0005 | 0.8150±0.0002 |
| ATKT | No | Yes | No | 0.7783±0.0005 | 0.8145±0.0002 |
| GKT | No | Yes | No | 0.7727±0.0006 | 0.8135±0.0004 |
| SAKT | No | Yes | No | 0.7693±0.0008 | 0.8124±0.0002 |
| SAINT | Yes | Yes | Yes | 0.8074±0.0007 | 0.8177±0.0006 |
| AKT | Yes | Yes | No | 0.8207±0.0008 | 0.8273±0.0007 |
| simpleKT | Yes | Yes | No | 0.8163±0.0006 | 0.8246±0.0005 |
| sparseKT | Yes | Yes | No | 0.8165±0.0015 | 0.8234±0.0009 |

We use the abbreviations to represent different information as follows: (1) **Q.ID**: the question ID; (2) **Q.Cont**: the question content, analysis content and question type of the corresponding question; (3) **KC ID**: the KC ID of the leaf node in KC route for the corresponding question; (4) **KC Cont**: the plaintext of the corresponding KC; and (5) **KC Route**: the KC route of the corresponding question, the weight of each level in the KC routes are learned by the model.

Table 3: The AUC and accuracy performance with auxiliary information on XES3G5M dataset.

| Model | Q.ID | KC ID | Q.Cont | KC Cont | KC Route | AUC | | | Accuracy | | |
|---|---|---|---|---|---|---|---|---|---|---|---|
| | | | | | | Add | Early Fusion | Late Fusion | Add | Early Fusion | Late Fusion |
| qDKT | Yes | No | No | No | No | 0.8225±0.0002 | - | - | 0.8301±0.0000 | - | - |
| | Yes | No | Yes | No | No | 0.8220±0.0004 | 0.8228±0.0005 | - | 0.8299±0.0002 | 0.8303±0.0003 | - |
| simpleKT | Yes | Yes | No | No | No | 0.8163±0.0006 | - | - | 0.8163±0.0006 | - | - |
| | Yes | Yes | Yes | Yes | No | 0.8251±0.0008 | 0.8277±0.0003 | 0.8251±0.0008 | 0.8303±0.0007 | 0.8313±0.0003 | 0.8303±0.0003 |
| | Yes | Yes | Yes | No | Yes | 0.8219±0.0006 | 0.8268±0.0012 | 0.8256±0.0006 | 0.8280±0.0007 | 0.8309±0.0003 | 0.8305±0.0003 |
| | Yes | Yes | Yes | Yes | Yes | 0.8245±0.0006 | 0.8262±0.0010 | 0.8227±0.0029 | 0.8302±0.0003 | 0.8304±0.0007 | 0.8294±0.0008 |
| AKT | Yes | Yes | No | No | No | 0.8207±0.0008 | - | - | 0.8273±0.0007 | - | - |
| | Yes | Yes | Yes | Yes | No | 0.8228±0.0004 | 0.8292±0.0007 | 0.8289±0.0008 | 0.8288±0.0003 | 0.8321±0.0005 | 0.8323±0.0004 |
| | Yes | Yes | Yes | No | Yes | 0.8216±0.0004 | 0.8287±0.0007 | 0.8295±0.0003 | 0.8285±0.0003 | 0.8319±0.0006 | 0.8327±0.0003 |
| | Yes | Yes | Yes | Yes | Yes | 0.8213±0.0003 | 0.8290±0.0007 | 0.8288±0.0009 | 0.8284±0.0004 | 0.8317±0.0006 | 0.8323±0.0005 |

To comprehensively evaluate the effects of auxiliary information, we select top performing methods from Table 2 in terms of AUC. Furthermore, since the IEKT has a reinforcement learning based sampling module to select students' individual cognition and acquisition representations, which is very time-consuming, we choose to use qDKT, simpleKT, and AKT as the representative approaches. Their AUC and accuracy results are shown in Table 3. We have the following three important observations described as follows: (1) augment auxiliary information in three fusion methods on simpleKT and AKT outperform the original models without auxiliary information. The simpleKT and AKT models with Q.Cont and KC Cont in the early fusion method have significant improvements of 1.14% and 0.85% in terms of the AUC scores; (2) with the comparison among three fusion methods for using the auxiliary information, the results of early fusion methods always perform best. In AKT, it gets the improvements of AUC performance in 0.64%, 0.71% and 0.77% compared to the results of add fusion way in three selection ways of question and KC information; (3) the AUC performance of using Q.Cont and KC Cont are the majority of the best performance in three fusion methods of using auxiliary information. After using more KC Route information, the performance declined. We suppose it is because the way of encoding KC Route is too simple that leads to the

misunderstanding of KT models. Due to the space limit, we provide the visualization comparison of question embeddings both with and without the KC Route information in the Appendix A.9; (4) the augmentation results of qDKT is slightly decline compared to the original qDKT without auxiliary information, it is because qDKT estimate students' knowledge states in question level, adding more information about KC make the model become harder to discriminate the individualized of questions.

## 4.5 Performance on Multi-step ahead Prediction

To make the KT prediction close to real application scenarios, we make a multi-step ahead prediction to students' future responses given their historical interaction sequence following [17], shown in Figure 4. The multi-step prediction will provide constructive feedback to learning path selection and construction for teachers, and help them flexibly prepare the teaching materials. The multi-step ahead prediction has two prediction methods including non-accumulative prediction and accumulative prediction. The non-accumulative prediction method aims to predict all future responses all at once by only using the given historical interactions. Besides the given historical interactions, the accumulative prediction approach also uses the last predicted value to predict the current response. Specifically, we vary the observed percentages of student historical interaction length from 20% to 90% with a step size of 10%. As discussed in Section 4.4, simply augmenting the auxiliary information also improves the performance of KT models on predicting students' future performance. Hence, to further investigate the effectiveness of auxiliary information augmentation in the multi-step ahead prediction setting, we keep the same representative DLKT approaches (qDKT, simpleKT and AKT) used in Section 4.4 with their corresponding best fusion approaches reported in Table 3, i.e., qDKT with Q.ID and KC ID, simpleKT with Q.ID, KC ID, Q.Cont and KC Cont in early fusion, AKT with Q.ID, KC ID and Q.Cont in late fusion as the representative approaches.

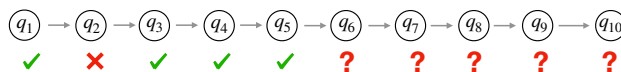

Figure 4: The graphical illustration of the multi-step ahead prediction problem.

From Figure 5, we can see that (1) the AUC and accuracy performance get higher results with the increasing percentage of historical interactions no matter whether augment DLKT models with auxiliary information or not, which matches the real-world educational scenario; (2) the majority of AUC and accuracy results of the models with auxiliary information (shown in solid lines) always outperform the baseline models without combining auxiliary information (shown in dotted lines), which indicates that with the limited historical interactions, the auxiliary information of questions provides more beneficial information to accurate the students' performance estimations; (3) the performance improvements of non-accumulative prediction with auxiliary information are higher than the results with accumulative setting compared the original baselines without auxiliary information respectively. This is because the non-accumulative prediction only considers the given historical interactions while the accumulative setting additionally uses the last prediction result to predict the next question, the limited information given leads to higher improvements from auxiliary information.

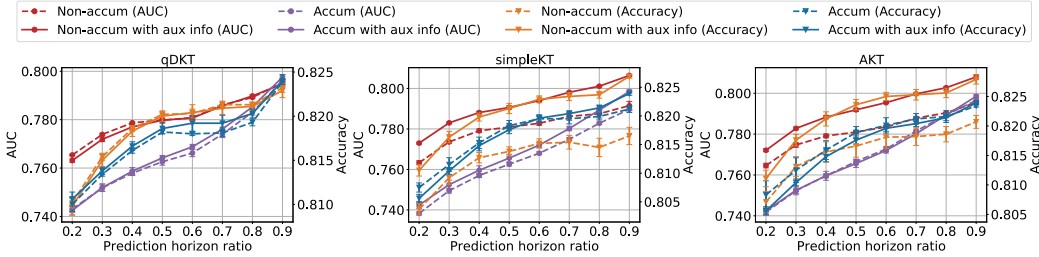

Figure 5: Accumulative (Accum) and Non-accumulative (Non-accum) predictions in the multi-step ahead scenario in terms of AUC and accuracy on XES3G5M. The results shown in dotted/solid lines are the AUC and accuracy results of the original qDKT/simpleKT/AKT without/with auxiliary information (e.g., Non-accum with aux info denotes the non-accumulative prediction of augmenting the qDKT/simpleKT/AKT with auxiliary information).

# 5   Limitation

In spite of the rich auxiliary information in XES3G5M is beneficial for predicting the future performance of students, it also has some limitations:

- **Data Imbalance**: There are two imbalance problems in XES3G5M:
  - **Question Type Imbalance.** From section 3, there are 80.27% fill-in-the-blank questions in XES3G5M while only 19.73% questions are multiple-choice.
  - **Interaction Imbalance.** The number of interactions of questions are ranging from 1 to 29,504 and the interaction numbers of KCs are ranging from 1 to 150,468. Since the large number of questions and KCs, there may be some questions and KCs have limited question-solving logs of students, which lead to the model being difficult to model students' behavior due to data sparsity.

- **Lack of Duration**: Due to the end timestamps of students' answers to each question were not recorded in the online learning platform, XES3G5M lacks duration for each interaction which is also helpful for estimating students' knowledge states over a series of time.

# 6   Conclusion and Future Work

In this paper, we develop a large-scale open source KT dataset named XES3G5M. The dataset has 5,549,635 interactions and about 18,066 students answered 7,652 questions associated with 865 leaf KCs. Notably, we provide the detailed information of question and knowledge side information, including question contents, question analysis, question types and the corresponding KC routes. Furthermore, we construct a DLKT benchmark on XES3G5M with 19 state-of-the-art DLKT baselines. To evaluate the effectiveness of the auxiliary information in XES3G5M, we conduct comprehensive experiments to leverage the auxiliary information to DLKT models. In the future, we will explore how to deeply model the relationship between KCs via graph neural networks based on the KC routes in XES3G5M and then improve the students' response prediction performance with the abundant information of questions.

# 7   Reproducibility Statement

To reproduce the experimental results, you can use the models in PYKT python library at `https://github.com/pykt-team/pykt-toolkit` and download the dataset of XES3G5M in `https://github.com/ai4ed/XES3G5M`. The details of data-preprocessing and the training hyper-parameters are shown in Section 3.2 and Section 4.1. Because the codes of models and dataset are all provided, all the results can be easily reproducible.

## Acknowledgments

This work was supported in part by National Key R&D Program of China, under Grant No. 2020AAA0104500; in part by Key Laboratory of Smart Education of Guangdong Higher Education Institutes, Jinan University (2022LSYS003); in part by National Joint Engineering Research Center of Network Security Detection and Protection Technology and in part by APRC - CityU New Research Initiatives (No.9610565, Start-up Grant for New Faculty of City University of Hong Kong), CityU - HKIDS Early Career Research Grant (No.9360163), Hong Kong ITC Innovation and Technology Fund Midstream Research Programme for Universities Project (No.ITS/034/22MS), Hong Kong Environmental and Conservation Fund (No. 88/2022), SIRG - CityU Strategic Interdisciplinary Research Grant (No.7020046, No.7020074).

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
