# A Appendix

## A.1 Dataset Accessibility and Long-Term Preservation Plan

The dataset can be freely downloaded from `https://github.com/ai4ed/XES3G5M`. The files of the dataset are stored on the Google Drive platform which can provide encrypted and secure access to the uploaded files and is widely used to share data with each other, researchers can directly download the dataset from `https://drive.google.com/file/d/1eFiIYyh5O2V90RAObrammGH6EpHvPDQe/view`. We believe this platform can ensure stable accessibility and our dataset can be stored for long-term preservation.

## A.2 License & Dataset Usage

Our dataset is opened sourced under the MIT license which can be found in `https://github.com/ai4ed/XES3G5M/blob/main/LICENSE`. Except the KT task defined in this paper, the users can use the dataset for other custom tasks under the license.

## A.3 Author Statement of Responsibility

The authors acknowledge full responsibility in the event of any rights violations and confirm the license associated with the dataset.

## A.4 Examples of Question Contents

Figures 6 and 7 show the examples of question contents in the XES3G5M.

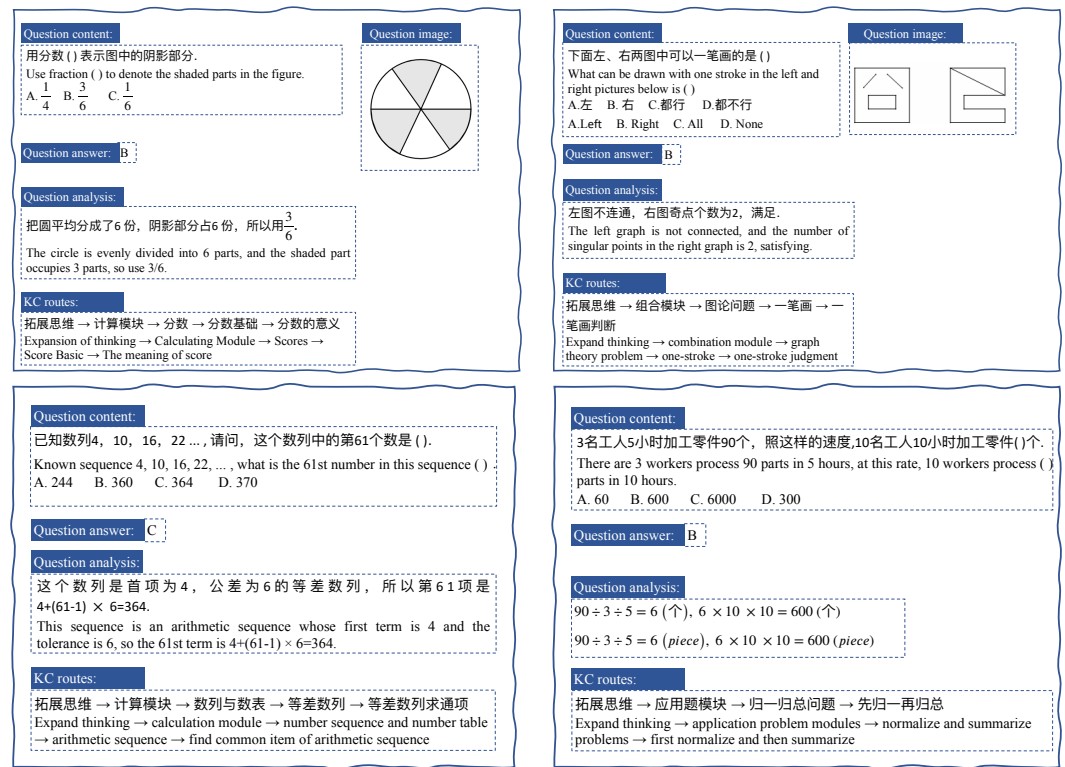

Figure 6: The examples of multi-choice questions.

## A.5 An Example of KC Routes

Figure 8 shows an example of KCs to illustrate the KC routes involved in the XES3G5M.

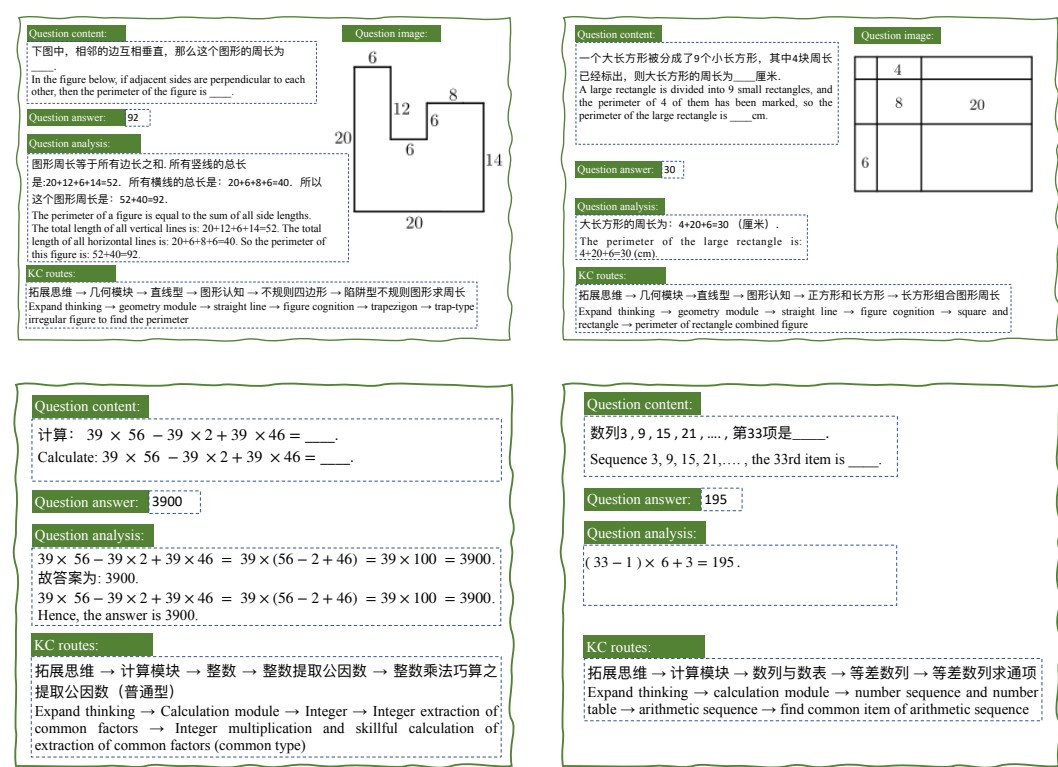

Figure 7: The examples of fill-in-the-blank questions.

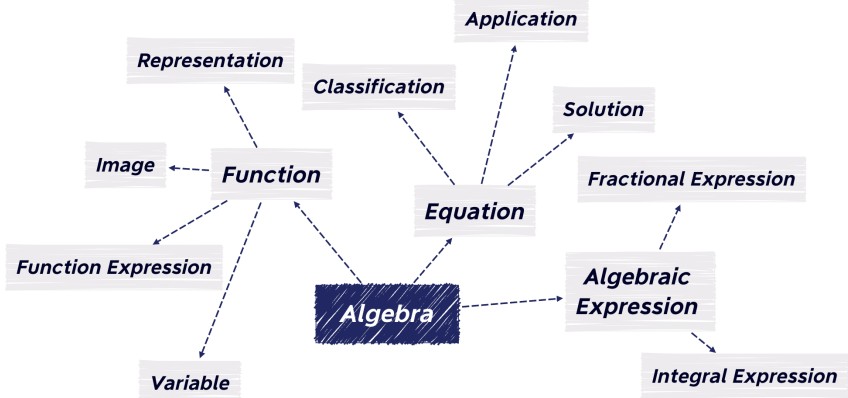

Figure 8: An example of KC routes.

## A.6 Dataset Structure and Storage Format

After preprocessing, we provide the files generated by pyKT[8] and the metadata files that involve auxiliary information. Specifically, the published data is stored in a data directory named XES3G5M. In the directory, there are the following files and folders:

- **kc_level**: the folder for the models which need KC level data to train and test:

  - **train_valid_sequences.csv**: this is the main data file for KT task to conduct offline model training and validation. Each student interaction sequence at question level is first expanded into KC level when a question is associated with multiple KCs. After that, each sequence is truncated into sub-sequences of length of 200. The data has been randomly split into 5 folds, the columns of this file are described as follows:
    * **fold**: the unique id of each fold, ranging from 0 to 4;
    * **uid**: the unique internal id of the user;
    * **questions**: the sequence of internal question ids;
    * **concepts**: the corresponding sequence of internal KC ids, in this file, we use only the leaf nodes of the KCs in the KC routes;
    * **responses**: the correctness of the students' answers, "1" means right, "0" means wrong;
    * **timestamps**: the regular start timestamps of the interactions, the timestamps are millisecond level;
    * **selectmasks**: if the length of current sequence is less than 200, we pad it to 200, "-1" indicates the corresponding interaction is padded or only used as history which needs to be ignored when calculating the loss and evaluating the models' performance;
    * **is_repeat**: "1" indicates that the current KC and its previous KC belong to the same question, "0" means this is a new question.

  - **test_question_window_sequences**: the test data file for question level prediction. Each students' sequence is truncated into 200 length. This data file is used to predict each KC for each question and aggregate the KC predictions in 4 different ways as mentioned in [17]. The columns in this file are consistent with them in file "train_valid_sequences.csv", the additional columns mean that:
    * **qidxs**: the interaction index sequence in the test dataset, each question at a particular timestamp has only one index;
    * **rest**: the rest KC counts of the interaction in the original sequences, it will be used in the predictions' fusion;
    * **orirow**: the corresponding row numbers of the interactions in the test dataset.

  - **test.csv**: this is the test data file to evaluate the KC level models' performance for multi-step ahead prediction scenario. Each row represents a test student interaction sequence. There are 3,613 student interaction sequences in this file. The columns in this file are same to file "test_question_window_sequences" except "cidxs":
    * **cidxs**: the interaction index sequence in the test dataset, the KCs from the same question at a particular timestamp have different indexes.

- **question_level**: the folder for the models which need question level data to train and test:

  - **train_valid_sequences_quelevel.csv**: this is the data file for KT models which need question level data to train. The columns are the same as "train_valid_sequences.csv", also note that in **concepts** column, "_" is used to split KCs of questions that have multi KCs;
  - **test_window_sequences_quelevel.csv**: the question level data file for KT models to predict their performance. It has the same columns with "train_valid_sequences_quelevel.csv";
  - **test_quelevel.csv**: this is the original test data file to evaluate the question level models' performance for multi-step ahead prediction scenario. The columns are consistent with them in file "train_valid_sequences_quelevel.csv".

- **metadata**: the folder of the metadata for questions:

  - **questions.json**: this file contains the detailed information of questions, the format of the data is a dictionary, each question is a key-value item, the key is the question id, the value is also a dictionary, each item in the dictionary means that:

---

[8]https://github.com/pykt-team/pykt-toolkit

* **content**: the question textual content;
* **kc_routes**: the textual of the KC routes;
* **answer**: the right answer of the question;
* **options**: the options of the question if the type of the question is multi-choice, when it is a fill-in-the-blank question, the value of this key is null;
* **analysis**: the textual content of the detailed analysis for resolving the question;
* **type**: the question type. 0: fill-in-the-blank question; 1: multi-choice question.
– **kc_routes_map.json**: this is the mapping file between the indexes of all KCs and their corresponding textual;
– **embeddings**: we additionally provide the embedding files of our questions, analysis and KCs. Specifically, we further pretrain the large-scale language model RoBERTa via the exercise data from a K-12 online learning platform in China [13]. Then we obtain the semantic representations by averaging all the word-level representations of each question, analysis and KC. The embedding fold includes "qid2content_emb.json", "qid2analysis_emb.json", and "cid2content_emb.json" which are the embeddings of questions, analysis and KCs respectively. In each JSON file, each key is the question ID or KC ID, and the value is the corresponding embedding;
– **images**: this is the folder of images for all questions. The name of the image file is corresponding to the question, each question may have 0 or more images, the images' formats are "question_qid-image_index" in question contents and options, and "analysis_qid-image_index" in question analysis, at the same time, the links of the images in the contents are also replaced to the name.

## A.7 Question Distribution

Figure 9(a) and 9(b) give the distributions of the number of KCs and the accuracy of each question, it is clear that most questions have only one KC and 53.75% questions have more than 80% accuracy. We also analyze the question and analysis length in XES3G5M, the distributions are shown in figure 9(c) and 9(d).

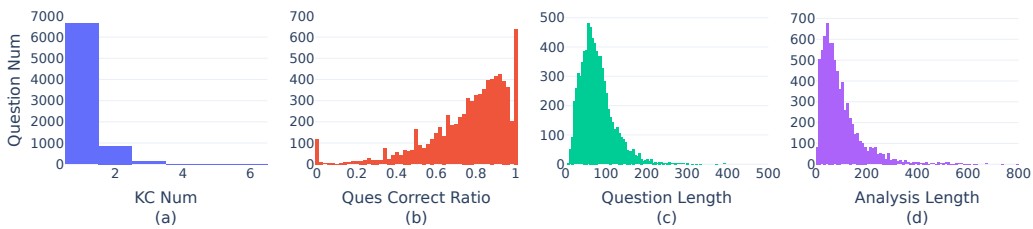

Figure 9: The distributions between questions and other information.

## A.8 Reproducibility Statement

The code for our benchmark of using the dataset can be found at `https://github.com/pykt-team/pykt-toolkit`, with the dataset download link `https://github.com/ai4ed/XES3G5M` and the training details in Section 4.1, the experimental results can be easily reproduced.

## A.9 Visualization of Question Embeddings

To examine the effectiveness of the learned representations in different fusion types, we conduct qualitative analysis on the question representations in simpleKT. We first select the top 10 KCs from the randomly selected 2000 sequences in the test file in XES3G5M. After that, we select the first 2000 questions in the selected sequences and get their corresponding representations of "Q.ID&KC ID&Q.Cont&KC Cont" and "Q.ID&KC ID&Q.Cont&KC Cont&KC Route" respectively. The visualizations are shown in Figure 10 and 11 in which different KCs are shown in different colors. From Figure 10, we can observe that after using the KC Route, although it provides richer information, the question embeddings of different KCs in add fusion type may overlap with each other, this makes the prediction performance drop which can be seen by comparing the corresponding

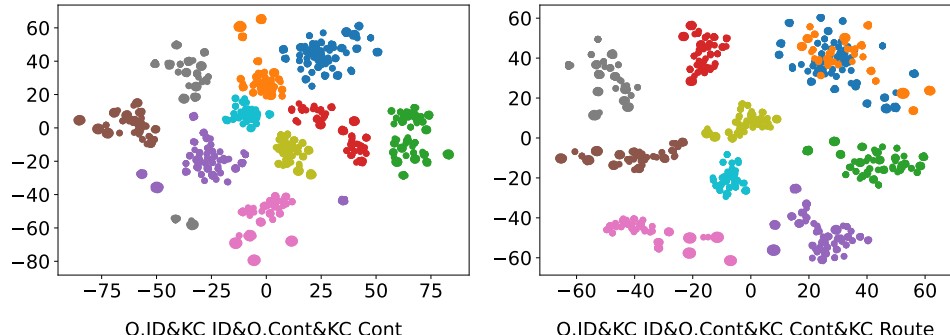

Figure 10: Visualization of question embeddings of add fusion for simpleKT in XES3G5M. Best viewed zoomed in and in color.

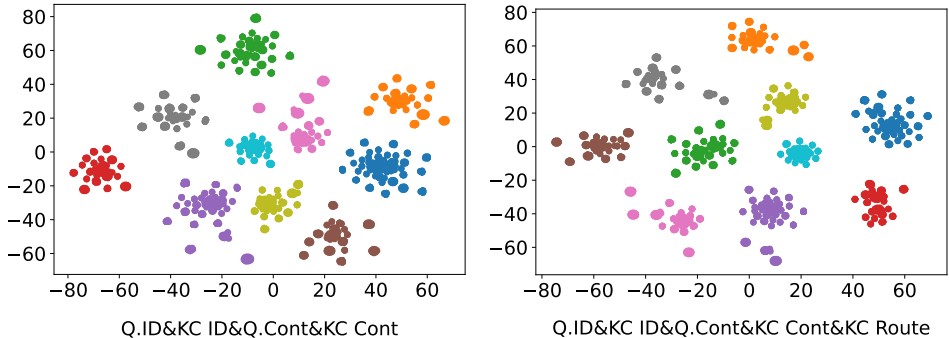

Figure 11: Visualization of question embeddings of early fusion for simpleKT in XES3G5M. Best viewed zoomed in and in color.

AUC in simpleKT and AKT in Table 3. In Figure 11, compared the question embeddings in the same KCs in early fusion, when using KC Route, the question embeddings are more similar than not used, we think the reason is that after using KC Route, the distinctiveness between questions has weakened in the same KCs. Those phenomenon indicate that the way of using KC Route needs more exploration in the future, and our simple usage methods can not leverage this information very well.

# B    XES3G5M Datasheet

To clearly provide a document about our XES3G5M, we follow the suggestions of the Datasheets [9] for Datasets to mainly answer the questions about the motivation, composition, collection process, preprocessing/cleaning/labeling, uses, distribution and maintenance of the dataset.

## B.1    Motivation

- **For what purpose was the dataset created?** *Was there a specific task in mind? Was there a specific gap that needed to be filled?*

  The XES3G5M dataset is to provide students' question-solving logs with rich auxiliary information to facilitate knowledge tracing research work which is limited by only anonymized ID-like features included in most of the existing public KT datasets.

- **Who created the dataset (e.g., which team, research group) and on behalf of which entity (e.g., company, institution, organization)?**

  The XES3G5M dataset is created from a real-world online math learning platform that is developed by TAL Education Group (TAL), one of the largest educational technology companies in China.

- **Who funded the creation of the dataset?** *If there is an associated grant, please provide the name of the grantor and the grant name and number.*

---

[9]https://arxiv.org/abs/1803.09010

This work was supported in part by National Key R&D Program of China, under Grant No. 2022YFC3303600; in part by Beijing Nova Program (Z201100006820068) from Beijing Municipal Science & Technology Commission; in part by Key Laboratory of Smart Education of Guangdong Higher Education Institutes, Jinan University (2022LSYS003) and in part by National Joint Engineering Research Center of Network Security Detection and Protection Technology.

- **Any other comments?**

  No.

## B.2 Composition

- **What do the instances that comprise the dataset represent (e.g., documents, photos, people, countries)?** *Are there multiple types of instances (e.g., movies, users, and ratings; people and interactions be- tween them; nodes and edges)?*

  The XES3G5M dataset consists of lots of student interactions, each interaction is associated question information, KC information, answer state and timestamps.

- **How many instances are there in total (of each type, if appropriate)?**

  The dataset contains 7,652 questions, and 865 KCs with 5,549,635 interactions from 18,066 students.

- **Does the dataset contain all possible instances or is it a sample (not necessarily random) of instances from a larger set?** *If the dataset is a sample, then what is the larger set? Is the sample representative of the larger set (e.g., geographic coverage)? If so, please describe how this representativeness was validated/verified. If it is not representative of the larger set, please describe why not (e.g., to cover a more diverse range of instances, because instances were withheld or unavailable).*

  All the question-solving logs in the dataset are answered by third-grade students, and all the questions are math questions. It is a representative subset of the learning status of third-grade students in math.

- **What data does each instance consist of?** *"Raw" data (e.g., unprocessed text or images) or features? In either case, please provide a description.*

  Each instance in the XES3G5M dataset is a student's question-solving log including student IDs, question IDs, answers (correct or incorrect), timestamps, question contents, KCs associated with the questions, and question types.

- **Is there a label or target associated with each instance?** *If so, please provide a description.*

  The target of the knowledge tracing task on XES3G5M is to predict whether the student can answer the future question correctly or not (1/0 denotes correct/incorrect respectively).

- **Is any information missing from individual instances?** *If so, please provide a description, explaining why this information is missing (e.g., because it was unavailable). This does not include intentionally removed information, but might include, e.g., redacted text.*

  Due to the end timestamps of students' answers to each question were not recorded in the online learning platform, XES3G5M lacks duration for each interaction which is also helpful for estimating students' knowledge states over a series of time.

- **Are relationships between individual instances made explicit (e.g., users' movie ratings, social network links)?** *If so, please describe how these relationships are made explicit.*

  We explicit all historical interactions of students via cluster them via student ID. Among the interactions of a student, we sort them via the timestamps to indicate the order of a student's answer sequence.

- **Are there recommended data splits (e.g., training, development/validation, testing)?** *If so, please provide a description of these splits, explaining the rationale behind them.*

  We perform standard 5-fold cross validation for all the combinations of pairs of method and dataset. We use 80% of student sequences for training and validation and use the rest 20% of student sequences for model evaluation.

- **Are there any errors, sources of noise, or redundancies in the dataset?** *If so, please provide a description.*

Since some samples in our dataset are collected by a series of auxiliary educational tools of TAL such as personalized learning applications, there will be some data duplication such as a question-solving log being recorded repeatedly and some data missing like missing and incomplete question content and associated KCs.

- **Is the dataset self-contained, or does it link to or otherwise rely on external resources (e.g., websites, tweets, other datasets)?** *If it links to or relies on external resources, a) are there guarantees that they will exist, and remain constant, over time; b) are there official archival versions of the complete dataset (i.e., including the external resources as they existed at the time the dataset was created); c) are there any restrictions (e.g., licenses, fees) associated with any of the external resources that might apply to a dataset consumer? Please provide descriptions of all external resources and any restrictions associated with them, as well as links or other access points, as appropriate.*

  The dataset is self-contained.

- **Does the dataset contain data that might be considered confidential (e.g., data that is protected by legal privilege or by doctor-patient confidentiality, data that includes the content of individuals' non-public communications)?** *If so, please provide a description.*

  Among the selected information, the student IDs and question IDs have the potential to reveal the identity of students and questions. Therefore, to better protect students' privacy, we have employed a lightweight data encryption method based on data mapping. For each student ID and question ID, we map it into a non-reversible digital identifier. The mapping identifier generates unique IDs in the dataset to guarantee each student and question has a unique identifier to avoid any duplication or confusion. The encryption method enables data security meanwhile improving data processing efficiency.

- **Does the dataset contain data that, if viewed directly, might be offensive, insulting, threatening, or might otherwise cause anxiety?** *If so, please describe why.*

  No.

- **Does the dataset identify any subpopulations (e.g., by age, gender)?** *If so, please describe how these subpopulations are identified and provide a description of their respective distributions within the dataset.*

  Our dataset only focuses on the question-solving logs of third-grade students.

- **Is it possible to identify individuals (i.e., one or more natural persons), either directly or indirectly (i.e., in combination with other data) from the dataset?** *If so, please describe how.*

  No, to better protect students' privacy, we have employed a lightweight data encryption method based on data mapping. For each student ID, we map it into a non-reversible digital identifier.

- **Does the dataset contain data that might be considered sensitive in any way (e.g., data that reveals race or ethnic origins, sexual orientations, religious beliefs, political opinions or union memberships, or locations; financial or health data; biometric or genetic data; forms of government identification, such as social security numbers; criminal history)?** *If so, please provide a description.*

  No.

- **Any other comments?**

  No.

### B.3 Collection Process

- **How was the data associated with each instance acquired?** *Was the data directly observable (e.g., raw text, movie ratings), reported by subjects (e.g., survey responses), or indirectly inferred/derived from other data (e.g., part-of-speech tags, model-based guesses for age or language)? If the data was reported by subjects or indirectly inferred/derived from other data, was the data validated/verified? If so, please describe how.*

  We mainly collect data from two perspectives, including questions designed by teachers during class and homework assignments. First, for the questions in class, we collect educational data from an online learning system that is developed by TAL Education Group (TAL), one of the largest educational technology companies in China. For each sample, we collect question textual information and the corresponding student's answer. Second, for the questions of homework

assignments, we utilize a series of auxiliary educational tools of TAL such as personalized learning applications to collect students' question-solving logs in their homework.

- **What mechanisms or procedures were used to collect the data (e.g., hardware apparatuses or sensors, manual human curation, software programs, software APIs)?**
  How were these mechanisms or procedures validated? We utilize a series of auxiliary educational tools of TAL such as personalized learning applications to collect students' question-solving logs.

- **If the dataset is a sample from a larger set, what was the sampling strategy (e.g., deterministic, probabilistic with specific sampling probabilities)?**
  Not applicable.

- **Who was involved in the data collection process (e.g., students, crowdworkers, contractors) and how were they compensated (e.g., how much were crowdworkers paid)?**
  The question-solving logs are automatically collected by the online learning platform and auxiliary educational tools of TAL, which is not required by crowdworkers.

- **Over what timeframe was the data collected?** *Does this timeframe match the creation timeframe of the data associated with the instances (e.g., recent crawl of old news articles)? If not, please describe the timeframe in which the data associated with the instances was created.*
  The record times of the interactions in our dataset are from 2018 to 2021.

- **Were any ethical review processes conducted (e.g., by an institutional review board)?** *If so, please provide a description of these review processes, including the outcomes, as well as a link or other access point to any supporting documentation.*
  No.

### B.4 Preprocessing/Cleaning/Labeling

- **Was Any Preprocessing/cleaning/labeling of the data done(e.g.,discretization or bucketing, tokenization, part-of-speech tagging, SIFT feature extraction, removal of instances, processing of missing values)?** *If so, please provide a description. If not, you may skip the remaining questions in this section.*
  We perform data preprocessing with the following steps: (1) we remove the questions with missing and incomplete content; (2) we remove the questions that miss the associated KCs; (3) the questions in the dataset are only in single choice and fill-in-the-blank types.

- **Was the "raw" data saved in addition to the preprocessed/cleaned/labeled data (e.g., to support unanticipated future uses)?** *If so, please provide a link or other access point to the "raw" data.*
  We release our dataset with "raw" data and preprocessed/cleaned/data both at `https://github.com/ai4ed/XES3G5M`.

- **Is the software that was used to preprocess/clean/label the data available?** *If so, please provide a link or other access point.*
  No.

- **Any other comments?**
  No.

### B.5 Uses

- **Has the dataset been used for any tasks already?** *If so, please provide a description.*
  The dataset has been conducted on knowledge tracing task, as seen in Section The XES3G5M Benchmark.

- **Is there a repository that links to any or all papers or systems that use the dataset?** *If so, please provide a link or other access point.*
  Yes, the DLKT models used in our paper can be found in PYKT python library at `https://github.com/pykt-team/pykt-toolkit` and the dataset is available at `https://github.com/ai4ed/XES3G5M`.

- **What (other) tasks could the dataset be used for?**
  The users can use the dataset for other custom tasks such as exercise recommendations.

- **Is there anything about the composition of the dataset or the way it was collected and preprocessed/cleaned/labeled that might impact future uses?** *For example, is there anything that a dataset consumer might need to know to avoid uses that could result in unfair treatment of individuals or groups (e.g., stereotyping, quality of service issues) or other risks or harms (e.g., legal risks, financial harms)? If so, please provide a description. Is there anything a dataset consumer could do to mitigate these risks or harms?*

  No.

- **Are there tasks for which the dataset should not be used?** *If so, please provide a description.*

  No.

- **Any other comments?**

  No.

## B.6 Distribution

- **Will the dataset be distributed to third parties outside of the entity (e.g., company, institution, organization) on behalf of which the dataset was created?** *If so, please provide a description.*

  Yes. The dataset is freely available.

- **How will the dataset will be distributed (e.g., tarball on website, API, GitHub)?** *Does the dataset have a digital object identifier (DOI)?*

  The dataset can be freely downloaded from `https://github.com/ai4ed/XES3G5M`.

- **When will the dataset be distributed?**

  The dataset is ready.

- **Will the dataset be distributed under a copyright or other intellectual property (IP) license, and/or under applicable terms of use (ToU)?** *If so, please describe this license and/or ToU, and provide a link or other access point to, or otherwise reproduce, any relevant licensing terms or ToU, as well as any fees associated with these restrictions.*

  Our dataset is open sourced under the MIT license which can be found in `https://github.com/ai4ed/XES3G5M/blob/main/LICENSE`.

- **Have any third parties imposed IP-based or other restrictions on the data associated with the instances?** *If so, please describe these restrictions, and provide a link or other access point to, or otherwise reproduce, any relevant licensing terms, as well as any fees associated with these restrictions.*

  No.

- **Do any export controls or other regulatory restrictions apply to the dataset or to individual instances?** *If so, please describe these restrictions, and provide a link or other access point to, or otherwise reproduce, any supporting documentation.*

  No.

- **Any other comments?**

  No.

## B.7 Maintenance

- **Who will be supporting/hosting/maintaining the dataset?** *If so, please provide a description.*

  The dataset will be supported/hosted/maintained by TAL.

- **How can the owner/curator/manager of the dataset be contacted (e.g., email address)?**

  The dataset maintainers can be contacted via issues or pull requests on the XES3G5M Github repository.

- **Is there an erratum?** *If so, please provide a link or other access point.*

  No.

- **Will the dataset be updated (e.g.,to correct labeling errors,add new instances,delete instances)?** *If so, please describe how often, by whom, and how updates will be communicated to dataset consumers (e.g., mailing list, GitHub)?*

  The dataset is complete and will not be updated.

- **If the dataset relates to people, are there applicable limits on the retention of the data associated with the instances (e.g., were the individuals in question told that their data would be retained for a fixed period of time and then deleted)?** *If so, please describe these limits and explain how they will be enforced.*

  No.

- **Will older versions of the dataset continue to be supported/hosted/maintained?** *If so, please describe how. If not, please describe how its obsolescence will be communicated to dataset consumers.*

  There is no older version of the dataset.

- **If others want to extend/augment/build on/contribute to the dataset, is there a mechanism for them to do so?** *If so, please provide a description. Will these contributions be validated/verified? If so, please describe how. If not, why not? Is there a process for communicating/distributing these contributions to dataset consumers? If so, please provide a description.*

  Any modification and extension of the dataset under the MIT license is permitted.

- **Any other comments?**

  No.