# OpenReview forum: "XES3G5M: A Knowledge Tracing Benchmark Dataset with Auxiliary Information"
_NeurIPS.cc/2023/Track/Datasets_and_Benchmarks — NeurIPS 2023 Datasets and Benchmarks Poster_

### Official Review · Reviewer_wWbZ · 2023-06-24
**XES3G5M: A Knowledge Tracing Benchmark Dataset with Auxiliary Information**

**Rating:** 5
**Confidence:** 3

**Strengths:**

Rich Auxiliary Information: The XES3G5M dataset provides extensive auxiliary information, including question textual content, question types, question analysis, KC routes, and question answers. This rich information enhances the representation and understanding of questions and enables more effective knowledge tracing.

Large-Scale and Real-World Data: The dataset contains a substantial number of interactions from a diverse set of students, making it suitable for training and evaluating KT models. The inclusion of real-world data enhances the dataset's applicability to practical educational scenarios.

Comprehensive Benchmark: The paper conducts a comprehensive benchmark of 15 DLKT models on the XES3G5M dataset, providing a thorough evaluation of the models' performance and demonstrating the benefits of leveraging the auxiliary information.


**Additional Feedback:**

Providing examples from the XES3G5M dataset, such as question-answer pairs, question analysis, and KC routes, would help readers better understand the richness of the auxiliary information.

Further discussion on the implications of data imbalances and strategies to address them would enhance the practical applicability of the dataset.

Exploring potential correlations between the auxiliary information and student performance could provide valuable insights into the effectiveness of leveraging such information for knowledge tracing.

**Clarity:**

The paper is well-written and effectively conveys the motivation, contributions, and methodology. The description of the dataset collection and preprocessing steps provides clarity on the data preparation process.

**Correctness:**

The claims made in the paper appear to be accurate based on the information provided. The dataset construction process and the incorporation of auxiliary information are described in detail, lending credibility to the proposed methodology.

**Documentation:**

The paper provides sufficient information on data collection, preprocessing steps, and the structure of the dataset. However, more details on data availability, maintenance, and ethical considerations would be valuable for potential users of the dataset.

**Ethics:**

No.

**Limitations:**

Yes.

**Opportunities For Improvement:**

Imbalanced Data: The dataset exhibits imbalances in question types and interaction frequencies. The high proportion of fill-in-the-blank questions and varying interaction numbers for questions and KCs may pose challenges due to data sparsity and difficulty in modeling student behavior. Further discussion on the impact of these imbalances and potential mitigation strategies would be beneficial.

Lack of Duration Information: The absence of duration information for each interaction limits the ability to estimate students' knowledge states over time. The inclusion of duration data could provide valuable insights into the temporal aspects of knowledge tracing.

**Relation To Prior Work:**

The paper adequately discusses the limitations of existing KT datasets and highlights the novelty and uniqueness of the XES3G5M dataset, which contains rich auxiliary information. The benchmarking of state-of-the-art DLKT models demonstrates the advancement over previous contributions. However, further comparisons and discussions with related work would enhance the paper's context.

**Summary And Contributions:**

The paper presents XES3G5M, a large-scale dataset for knowledge tracing (KT) that incorporates rich auxiliary information about questions and their associated knowledge components (KCs). The dataset is collected from a real-world online math learning platform and contains 7,652 questions, 865 KCs, and 5,549,635 interactions from 18,066 students. The XES3G5M dataset provides extensive contextual information, including tree-structured KC relations, question types, textual contents, analysis, and student response timestamps. The paper also presents a comprehensive benchmark on 15 state-of-the-art deep learning-based knowledge tracing (DLKT) models and demonstrates the effectiveness of leveraging the auxiliary information in the dataset.

---

> ### Author Response · Authors · 2023-08-21
> **Thanks for the comments and appreciation of our work.**
>
> Thanks for the comments and appreciation of our work.
>
> Q1: Question about imbalanced Data.
>
> A1: Thank you for raising the issue of imbalanced data. This limitation was highlighted in our paper (lines 349-355). Real-world classroom dynamics and pedagogical practices often lead to certain question formats being favored, or certain KCs being given more emphasis than others. While this imbalanced data offers a realistic glimpse into student interactions, it also brings forth certain challenges. As you correctly pointed out, data sparsity issues and complexities in modeling student behavior might arise due to this skewed representation. In the future, we will delve deeper into the implications of these imbalances. We will also explore potential mitigation strategies, including data augmentation techniques, synthetic data generation, or specific modeling approaches aimed at addressing data sparsity and effectively capturing student behaviors.
>
> Q2: Lack of Duration Information.
>
> A2: Thank you for highlighting the importance of duration information in understanding students' knowledge states over time. While our XES3G5M dataset offers more auxiliary information compared to other current datasets, as we mentioned in lines 356-358 of our paper, it indeed does not include duration information. We have recognized the importance of duration information, and taking this opportunity, we would like to share some recent updates with you: our technical team has completed the upgrade of our online education platform, and in the next version of our dataset, we will provide even more auxiliary information, including duration information.
>
> Q3: Add more documents about data availability, and maintenance.
>
> A3: Our dataset is readily available to researchers who are interested, and it can be conveniently accessed via our GitHub repository: https://github.com/ai4ed/XES3G5M.
>
> Regarding maintenance, we have used the discussion module in github.com to keep answering any question related to our dataset, see https://github.com/pykt-team/pykt-toolkit/discussions. Meanwhile, our team will be responsible for ensuring the upkeep of the dataset, and it's worth noting that this dataset is regarded as final and will not undergo any further updates.
>
>
> Q4: Provide examples from the XES3G5M dataset.
>
> A4: Thanks for your suggestion. We have added more examples in Figure 6 and Figure 7 in Appendix A4.

---

> ### Author Response · Authors · 2023-08-30
> **Looking forward to your feedback**
>
> We sincerely appreciate your valuable comments on our work. We have tried our best to address the concerns. Is there any unclear point that we should/could further clarify?

---

### Official Review · Reviewer_adSw · 2023-07-14
**Review of paper 101 (new knowledge tracing dataset and benchmarks)**

**Rating:** 7
**Confidence:** 4

**Strengths:**

The main strengths of the paper are: (1) a new dataset with some unique, interesting metadata related to the hierarchical structure of knowledge components, which is often absent from similar datasets, and (2) a wide-ranging evaluation of common deep knowledge tracing methods applied to the dataset, which sets expectations for potential future research using this dataset.


**Additional Feedback:**

I would like to thank the authors for submitting this interesting work. I was especially intrigued by the lack of meaningful difference between most of the different knowledge tracing methods compared, suggesting perhaps radically different approaches are needed to really improve on the goals of knowledge tracing.


**Clarity:**

The clarity of the paper could be much improved. There were several portions I did not fully understand, including key details. For example, the data preprocessing section (3.2) does not make it clear whether preprocessing was applied to the dataset that is available, or whether preprocessing was done *before* making the dataset available. This is important to know for future researchers working with the data, and to assess potential selection bias issues.


**Correctness:**

The claims and results appear correct, and the paper makes use of existing benchmark tools which lends support to the validity of the machine learning results.


**Documentation:**

Yes, all details are present.


**Ethics:**

The checklist at the end says that consent was discussed, but it doesn't seem to be in the paper. Additionally, these are data collected from human subjects so it is questionable if all the human subjects questions are really "N/A" or not.


**Limitations:**

The paper describes an anonymization process for the dataset that is likely sufficient to prevent reidentification, since without information about students themselves, knowledge tracing data are rather simple and not highly student-specific.


**Opportunities For Improvement:**

One opportunity for improvement is in representativeness. The dataset presented represents a narrow age range (only 3rd grade). Research conducted using this dataset might have limited generalizability to math concepts at other levels, where math concept complexity may be higher and factors like students' reading ability become more or less important to predicting success.


**Relation To Prior Work:**

Yes, it is clearly different from prior work. Though not hugely novel, as this is a crowded area of research with many available datasets, the paper clarifies which aspects are unique in this one.


**Summary And Contributions:**

This paper describes a new knowledge tracing dataset. Knowledge tracing is a common task in educational data mining, where it is used to estimate student mastery of math concepts to assess progress and sequence curriculum appropriately. This paper describes a new dataset with more metadata than most knowledge tracing datasets, which may be of value to other researchers.

---

> ### Author Response · Authors · 2023-08-21
> **Thanks for the comments and appreciation of our work.**
>
> Thanks for the comments and appreciation of our work.  Your comment of "I would like to thank the authors for submitting this interesting work." is very encouraging to us.
>
> Q1: Question about representativeness.
>
> A1: Thank you for the insightful observation regarding the dataset's representativeness. You're right; different grade levels bring different complexities in mathematical concepts, and the demands on students' reading abilities may vary considerably.
>  While our dataset focuses primarily on the 3rd-grade age range, its strength lies in the richness of the auxiliary information it provides, as opposed to merely ID features of questions, KCs, and timestamps which most datasets offer. As mentioned in our paper (lines 86-88), "All aforementioned datasets mainly contain the ID features of questions and KCs and timestamps but lack information related to educational contexts that may be potentially useful in assessing students’ knowledge states." Our dataset, XES3G5M, provides comprehensive auxiliary information from a wide variety of educational scenes. We believe this unique attribute allows researchers to incorporate additional educational context into Deep Learning Knowledge Tracing (DLKT) models, potentially enhancing their performance and utility.
>
> Q2: Question about ethics concerns.
>
> A2: Thank you for raising this important concern regarding the checklist and the consent discussion. To clarify, our dataset is derived from automatic data collection through our system. We did not employ crowdsourcing, nor did we require subjects to manually label data for us. The nature of our data collection ensures that individual interactions are recorded in an aggregate and anonymized manner, without any identifiable information that could be linked back to a specific student. To address this, we have undertaken rigorous measures to obfuscate student identifiers. Through an anonymization process, we've ensured that the data cannot be mapped back to individual students. This anonymization is not just limited to student IDs but extends to any data that could potentially be used to identify a student, making re-identification practically impossible. Furthermore, the legal team at TAL Education Group has double checked the Chinese law policy that it is okay to release such dataset.
>
> Q3: More details about data preprocessing.
>
> A3: Thank you for your feedback regarding the clarity of the paper. We acknowledge that there might have been ambiguity in some sections, and we appreciate you pointing it out. To answer your questions, we follow the existing pre-processing steps suggested in https://pykt.org/ and the XES3G5M dataset is the one after pre-processing. We have re-written part of the texts between lines 141 - lines 146 to make it clear.

---

> > ### Comment · Reviewer_adSw · 2023-08-21
> >
> > Thank you for your clarifications to my comments, which were fairly minor. I think this could be a valuable contribution!

---

### Official Review · Reviewer_7CjA · 2023-07-21
**Interesting dataset**

**Rating:** 7
**Confidence:** 3
**Clarity:** Yes!

**Strengths:**

- The dataset is incredibly rich, and seems much larger than existing datasets in this space. It also appears significant that the authors are making available the underlying text of the questions, as well as information about the types of reasoning require for each question.
- Overall, the paper appears to be well written and clear. It does a reasonably good job of translating and describing a ML application that most generalist researchers would probably not be familiar with.


**Additional Feedback:**

Nothing else!

**Correctness:**

The results largely seem correct, there’s nothing that gives pause


**Documentation:**

Overall, seems good.


**Ethics:**

I think the salient concern is the release of student data. But, the paper has obfuscated IDs, and the data here is limited to performance on mathematical questions. I’m not sure how Chinese law governs dataset releases like this. It would be helpful if the authors could share a bit more about whether any compliance review was performed/necessary. I’m inclined to defer to them given that this also appears to be a partnership between the private sector and academia, and I’m imagining some type of due diligence was performed?

**Limitations:**

I think the limitations are appropriately addressed.


**Opportunities For Improvement:**

- One thing that’s lacking from both the main body and the appendix is a characterization of the dataset overall in terms of question content. The dataset captures K-12, but it’s not clear what the topical breakdown is? Is it fairly balanced across years, or is there a skew towards more beginner content?
- How many questions have images?
- Disaggregating performance by student characteristics might be helpful. For instance, in Table 2/3, how do the performance of the different models compare for students with <100 interactions, versus students with 100-200 interactions, etc? I could see this going in both directions–maybe students with more interactions are harder because these models can’t handle the long context.
- Section 4.1 is a bit unclear on what the inputs/outputs to these models are. What are the textual sequences being fed in? I’d potentially consider picking 5 exemplar models, describing these in more detail, and presenting their results in the main paper. The other 10 models can be provided in the Appendix. But right now there are so many models and I’m guessing most people aren’t familiar with them, so it’s a bit overwhelming. [I realize this is a significant writing change, so I’m not anchoring my reviewing score on this–it’s just some feedback that might be helpful]
- For the Section 4 results, are you using the embeddings you release?
- It might be nice to have more examples of questions in the Appendix.

Minor/Typos:
- L88: may potentially _be_ useful
- L92: [B]ased
- In Figure 2, could you mention in the description that the questions have been translated for clarity?


**Relation To Prior Work:**

My suspicion (and please correct me if I’m wrong!) is that there’s significant variance across countries when it comes to educational progress/pathways/outcomes, due to different pedagogical approaches and other factors. Maybe it would be useful to include the country-of-origin in Table 1?


**Summary And Contributions:**

- Knowledge tracing (KT) is a task that predicts students’ future performance based on their historical learning interactions
- Previous datasets for KT only reference questions and additional knowledge components by ID, thereby limiting the effectiveness of models and capping the types of methods researchers can explore
- The paper makes several contributions. First, it presents a large scale dataset consisting of studying interaction logs collected from a K-12 online learning platform in China. Second, it conducts a systematic analysis of 15 state-of-the-art DLKT models on the dataset. Third, it shows that leveraging the auxiliary information in DLKT models is important for greater performance.

---

> ### Author Response · Authors · 2023-08-21
> **Thanks for the comments and appreciation of our work.**
>
> Thanks for the comments and appreciation of our work.  Your comments of "The dataset is incredibly rich, and seems much larger than existing datasets in this space. " and "It does a reasonably good job of translating and describing a ML application that most generalist researchers would probably not be familiar with." are very encouraging to us.
>
> Q1:  Absence of a comprehensive characterization of the dataset's question content.
>
> A1: Thank you for your insightful observation. In Figure 9 (c) and Figure 9 (d) in Appendix 7, we provide the length (number of words) distributions of  both question content and analysis content. Meanwhile, in Appendix 9, we offer a visualization of the embeddings of the questions. All the questions in our dataset is math question in grade 3 that are used in the past 5 years at TAL Education Group, which is one of the largest after-school tutoring companies in China.
>
> Q2: How many questions have images?
>
> A2: Thank you for raising this question about the presence of images in our dataset. To clarify:
>
> - Total questions containing images: 2,740
> - Number of images within the question content: 2,221
> - Number of images within the question analysis: 1,137
>
> In future versions, we'll ensure to provide such detailed statistics upfront to give readers a comprehensive overview of the dataset's attributes.
>
> Q3: More analysis from student characteristics, for example, interaction sequence length.
>
> A3: Thank you for your valuable feedback. We recognize the significance of analyzing performance based on student characteristics, particularly when considering the number of interactions. It provides a nuanced understanding of how different models perform for various student engagement levels.
>
> In this work, we only select the students' sequences with more than 200 interactions to better model students' learning processes with enough historical interaction information. This is mentioned at lines 139-141. Meanwhile, we have shown the interaction length distribution in Figure 3(a) that none of our sequence is less than 200, which typically a widely used threshold to determine long or short sequences. Therefore, all the student interaction sequences can be treated as long interaction sequence. We plan to study whether the model can capture the "extremely long" contexts, i.e., sequence length >=500, in the future.
>
>
> Q4: Question of "are you using the embeddings you release?"
>
> A4: Thank you for your query regarding Section 4. The answer is YES. The results presented in this section were indeed obtained using the embeddings we have released.
>
> Q5: Add more examples of questions.
>
> A5: Thanks for your suggestion. We have added more examples in Figure 6 and Figure 7 in Appendix A4.
>
> Q6: Add "questions have been translated" to Figure 2 for clarity.
>
> A6: Thank you for your suggestion.  The phrase "questions have been translated" has been added into Figure 2.
>
> Q7: Country of origin in Table 1.
>
> A7: Thank you for your perceptive insights. We agree that educational progress indeed demonstrates substantial variability across countries, owing to the diverse range of pedagogical methodologies, cultural impacts, and other influential factors. However, this difference vanishes as the students grow up.
>
> For example, different math questions might be used in elementary schools in US and China and there is no difference in graduate schools' exercises and the graduate schools may have international students as well.  For example, the Statics2011 dataset originates from an online engineering statics educational course held at Carnegie Mellon University during the Fall of 2011. This encompasses participation from multiple countries. Consequently, it is difficult to assign single country of origin to each dataset.
>
>
> Q8: Concern about the release of student data in Chinese.
>
> A8: Thank you for highlighting the paramount importance of data privacy, especially concerning student data. To address this, we have undertaken rigorous measures to obfuscate student identifiers. Through an anonymization process, we've ensured that the data cannot be mapped back to individual students. This anonymization is not just limited to student IDs but extends to any data that could potentially be used to identify a student, making re-identification practically impossible. Furthermore, the legal team at TAL Education Group has double checked the Chinese law policy that it is okay to release such dataset.

---

> > ### Comment · Reviewer_7CjA · 2023-08-21
> >
> > Thank you for your detailed comments! My review was fairly positive to begin with so I'll leave it unchanged. I hope to see this paper at the conference.

---

### Official Review · Reviewer_PA9g · 2023-07-23
**a large and comprehensive dataset**

**Rating:** 6
**Confidence:** 3
**Correctness:** The claims of the paper are sufficien…
**Clarity:** The paper is clear

**Strengths:**

The paper describes a large dataset which contains a lot more metadata that is relevant to the problem, and it seems reasonable to believe that this additional metadata enables a finer-grained analysis of students' learning path.

**Additional Feedback:**

see previous fields for my comments

**Documentation:**

The documentation is scarse as the dataset only contains raw files and the readme only points to he downlad link. The format seems fairly straightforward though, so I suspect that the high-level description provided in the paper is sufficient to use the dataset, even though I'm not sure of it.

**Ethics:**

The main potential flag here is on the privacy of the students, this is addressed in Section 3.3 where the student's name have been transformed to ids by cryptographic hash, which seems reasonable to me in that context.

**Limitations:**

the paper is clear about the data/technical limitations of the work.

In my view, what is unclear is how this data is used. If we want to use these models to generate alternative learning paths for students, as eluded to in the paper, then we need to evaluate counterfactuals -- one such counterfactual could be "what would be the student score if I gave them this lecture instead of this one", but it depends on how we can influence the student's learning process. It is unclear to me how this dataset could be used to model such counterfactuals.

**Opportunities For Improvement:**

In the evaluation, it is not clear what is predicted  by the models (the student answers? whether the student answers correctly?). I understand this follows a standard evaluation procedure in the domain of knowledge tracing, but it is unclear to a reader outside this community.

It is unclear what the scope of the research is. Here, I see the value to the knowledge tracing community, but it is unclear to me what the dataset bings to the broader NeurIPS community.

**Relation To Prior Work:**

There is sufficient references to prior work in the area of knowledge tracing and their public datasets

**Summary And Contributions:**

The paper presents a new dataset that contains interactions between students and exercices in online courses. The dataset is the largest dataset in terms of number of questions for math, is in Chinese where many of the other datasets are in English, and more importantly contains a wealth of metadata associated to the questions, as well as additional relations between "knowledge components", which form the basic description of knowledge acquisition.

The paper also includes an extensive evaluation of many deep learning models on tasks of predicting student responses, and show that the additional metadata helps improve predictive performance on these tasks.

---

> ### Author Response · Authors · 2023-08-21
> **Thanks for the comments and appreciation of our work.**
>
> Thanks for the comments and appreciation of our work.
>
> Q1: More details about evaluation.
>
> A1: Thank you for highlighting this point. To clarify, our model predicts whether a student answers a question correctly or not. In our paper, Figure 1 offers a visual representation of the Knowledge Tracing (KT) problem, and this specific detail is discussed in lines 21-22.
>
> Q2: Scope of the research and value for NeurIPS community.
>
> A2: Thank you for pointing out this concern. We do acknowledge the importance of illustrating the relevance of our dataset to the wider NeurIPS community.
>
> Please note that the first influential work called DKT that applies RNN/LSTM into the KT problem is published at NIPS 2015 [1], which has 1000+ citations. This work opens the new era of deep learning based knowledge tracing.
>
> Our dataset not only serves the knowledge tracing community but also aligns with the interests and pursuits of the NeurIPS community. For instance, there have been previous works presented at NeurIPS, such as pykt[2], "The NeurIPS 2020 Education Challenge"[3], and the "NeurIPS 2022 CausalML Challenge: Causal Insights for Learning Paths in Education"[4], which delve into similar areas. Given this context, we firmly believe that our XES3G5M dataset will offer value and insights to the NeurIPS audience.
>
> [1] Piech, Chris, et al. "Deep knowledge tracing." Advances in neural information processing systems 28 (2015).
>
> [2] Liu, Zitao, et al. "pyKT: a python library to benchmark deep learning based knowledge tracing models."Advances in Neural Information Processing Systems 35 (2022): 18542-18555.
>
> [3] https://competitions.codalab.org/competitions/25449
>
> [4] https://codalab.lisn.upsaclay.fr/competitions/5626
>
> Q3: Question about " It is unclear to me how this dataset could be used to model such counterfactuals".
>
> A3: Thank you for raising this valid concern. We apologise that this dataset is not able to support evaluations of counterfactuals. Such evaluations are very difficult and to the best of my knowledge, there is no existing offline static datasets can be used to model such counterfactuals.
>
> Q4: Documentation is scarse.
>
> A4: Thank you for pointing out the need for more comprehensive documentation. While our dataset format is intentionally kept simple for easy to use, we understand the importance of clear documentation. Detailed specifications of our data format can be found in the Appendix A6 (see lines 569 - 639). Furthermore, we have updated the README file to provide a thorough description, ensuring clarity and facilitating its use by the community.

---

> ### Author Response · Authors · 2023-08-30
> **Looking forward to your feedback**
>
> We sincerely appreciate your valuable comments on our work. We have tried our best to address the concerns. Is there any unclear point that we should/could further clarify?

---

### Official Review · Reviewer_vgkD · 2023-07-24
**A real-world KT dataset in Chinese following previous practices**

**Rating:** 6
**Confidence:** 2

**Strengths:**

1. This paper provides a large-scale education dataset that can be used to track students’ performance. It can be helpful for promoting a data-driven paradigm in modeling students’ learning processes. Math problem datasets also benefit from it.
2. This paper is well-written and clearly claims their contribution and limitation.
3. This paper conducts a comprehensive benchmark and presents how to use contextual information in their dataset to predict the future behavior of students.


**Additional Feedback:**

See opportunities

**Clarity:**

This paper conducts a new dataset and also makes a benchmark. The new dataset is the largest in the math Knowledge Tracing domain and provides rich auxiliary information to track students’ performance. And the benchmark performed well. But it will be better if this provides more recent models.


**Correctness:**

This dataset is sound since it comes from industrial paper. It also follows from existing practices.

**Documentation:**

This paper introduces the detail to collect data and how to avoid privacy problems. But data resource comes from the real-world platform and the availability and maintenance has problems.

**Ethics:**

The privacy information is properly discussed in the paper.

**Limitations:**

This paper adequately talks about the limitation of their dataset. For the imbalanced problems in question types, it seems to be better if conduct a sub-dataset to set the same scale of two question types. Then, this paper can vary if imbalanced problems have important effectiveness.

**Opportunities For Improvement:**

This paper is not only imbalanced in question types but also lacks useful question types such as Computational Questions, Proof Questions.
The deep learning model used in this paper doesn’t include the latest models.

**Relation To Prior Work:**

This paper talks about the difference with the previous dataset detailedly and list their strengths.

**Summary And Contributions:**

Summary: This paper collects a large knowledge-tracking dataset from the real-world platform, which contains math questions and students’ associated intersections. Also, this paper builds a benchmark on 15 SOTA deep learning-based knowledge tracing(DLKT) models and shows the performance of the models on how to use the semantic information.

Contributions:
1. This paper provides the largest math domain Knowledge Tracing dataset which has rich contextual information.
2. This paper conducts a systematic benchmark on their dataset based on 15 deep learning models and shows how to use multi-step ahead predictions in the Knowledge Tracing task.

---

> ### Author Response · Authors · 2023-08-21
> **Thank you for the detailed comments.**
>
> Thank you for the detailed comments. Below, we clarify your key concerns:
>
> Q1: Lack of useful question types.
>
> A1: Thank you for your constructive feedback. As we highlighted in the "Limitations" section, our dataset primarily focuses on multiple-choice and fill-in-the-blank question types. We intentionally filtered out some question types, such as MWP (Math word problems), because our system cannot automatically score them. Unlike multiple-choice and fill-in-the-blank items, Computational Questions and Proof Questions are subjective in nature, leading to students' responses not being presented in a standardized format. Currently, our system faces challenges in evaluating these diverse student responses automatically. However, as you rightly pointed out, these questions hold significant value. Hence, we are in the process of creating a high-quality dataset specifically for evaluating subjective questions to help scholars in the relevant fields to address these challenges.
>
> Q2: Add some recent models.
>
> A2: Thank you for your suggestion. The current experiments are based on representative baselines in 2020-2022. We have added experimental results from 4 newly published (2022-2023) DLKT approaches (DIMKT，AT-DKT，QIKT，sparseKT). Their experimental results are added in Table 2 and their descriptions can be found in Section 4.2.
>
> [DIMKT] from paper: Assessing Student’s Dynamic Knowledge State by Exploring the Question Difficulty Effect, SIGIR'2022.
>
> [AT-DKT] from paper: Enhancing Deep Knowledge Tracing with Auxiliary Tasks, WWW'2023.
>
> [QIKT] from paper: Improving Interpretability of Deep Sequential Knowledge Tracing Models with Question-centric Cognitive Representations, AAAI'2023.
>
> [sparseKT] from paper: Towards Robust Knowledge Tracing Models via k-Sparse Attention, SIGIR'2023.
>
>
> Q3: Using a sub-dataset with the same scale of two question types.
>
> A3: Thank you for your insightful feedback. Your suggestion of constructing a balanced sub-dataset to specifically examine the influence of this imbalance is well-taken. We appreciate this constructive advice and will certainly incorporate it into our future research to offer a more comprehensive perspective.
>
> Q4: The concern about the availability and maintenance of the datasets.
>
> A4: Thank you for addressing this important matter. Our dataset is readily available to researchers who are interested, and it can be conveniently accessed via our GitHub repository: https://github.com/ai4ed/XES3G5M. Regarding maintenance, we have used the discussion module in github.com to keep answering any question related to our dataset, see https://github.com/pykt-team/pykt-toolkit/discussions. Meanwhile, our team will be responsible for ensuring the upkeep of the dataset, and it's worth noting that this dataset is regarded as final and will not undergo any further updates.

---

> ### Author Response · Authors · 2023-08-30
> **Looking forward to your feedback**
>
> We sincerely appreciate your valuable comments on our work. We have tried our best to address the concerns. Is there any unclear point that we should/could further clarify?

---

### Decision · Program_Chairs · 2023-09-22

**Decision:**

Accept (Poster)

**Comment:**

The paper introduces a large-scale dataset of student interactions from a K-12 online learning platform in China. The paper evaluates 15 deep learning-based knowledge tracing (DLKT) models on the dataset and demonstrates that using meta data can improve their performance. Although, there are some concerns such as: limited question types, imbalanced interaction frequencies.
Overall, the paper provides a valuable dataset and contribution to the broader research community.